# DISTORTION REGULARIZATION FOR DISTANCE-PRESERVING GRAPH EMBEDDING

## ABSTRACT

Vertex embedding of graph-structured data offers the advantage of representing the graph in a low-dimensional continuous space, but it does not guarantee the preservation of distances. In this paper, we introduce a general distortion measure that can be integrated into loss functions. The distortion loss can be used as a regularization term, effectively maintaining pairwise distance relationships during embedding. We also show that Gaussian kernel embedding is a form of minimum-distortion embedding. Furthermore, we analyze and compare the strengths of different distortion measures through theoretical analysis. Finally, we demonstrate the effectiveness of distortion regularization across multiple downstream tasks using benchmark datasets. The results confirm that regularization based on distortion is effective and generally improves the performance of downstream tasks.

## 1 INTRODUCTION

With graph-structured data, the typical graph learning process first embeds the graph into a latent space, often with lower dimensions, and then applies machine learning algorithms on these embeddings for subsequent tasks. Creating high-quality graph representations in a continuous latent space is crucial for the performance of downstream tasks, especially when the encoder is trained in a self-supervised manner, and the downstream decoder only uses the learned representations without gradient flows propagating back to the encoder.

A powerful approach for scenarios with limited or no labeled data is contrastive learning (Zheng et al., 2022; Velickovic et al., 2019; Zhu et al., 2020; 2021; Thakoor et al., 2021; Xia et al., 2022; Lee et al., 2022). With the InfoNCE loss (van den Oord et al., 2018), contrastive learning effectively creates a self-supervised task that learns representations from unlabeled data, making it a cornerstone of modern self-supervised learning. However, avoiding representation collapse is a critical component of contrastive learning. Without addressing collapse, the self-supervised objective would fail. A variety of ad hoc techniques, e.g., diverse data augmentations (Zhu et al., 2021; Lee et al., 2022; Li et al., 2023), large negative sample sets (Xia et al., 2022; He et al., 2024b), or architectural constraints (Grill et al., 2020; He et al., 2024a), are employed to maintain representation diversity. The representation scattering mechanism proposed in He et al. (2024a) is a cost-effective approach to address representation collapse.

Representation scattering might be important for contrastive learning. However, it may not be necessary for other learning methods. A more general principle for graph learning, which applies to all encoders and decoders, is distance preservation—ensuring that the exact distance relationships between nodes in the original graph are maintained when representing them as vectors in a lower-dimensional space. Mathematically, a distance-preserving embedding $f$ aims to ensure that $d_H(f(x_i), f(x_j)) \approx g(d_S(x_i, x_j))$, where $g$ is some function relating sample space distances $d_S$ to embedding space distances $d_H$.

The concept of distance preservation, independent of specific distance measures in the graph and embedding spaces, is crucial for ensuring that the learned embeddings effectively capture the graph's topological properties, such as proximity or connectivity, as well as patterns of nodal features, such as similarity or directional alignment. What specific distance measures should be used in the sample space and the embedding space depends on the tasks at hand. Along with appropriate distance measures, distance-preserved embeddings of a lower dimension can capture both structural and

feature-based patterns, allowing not only improved efficiency but also improved performance for downstream tasks.

To measure how well an encoder preserves the distances in the sample space, a concept called "distortion" initially emerged in the field of geometry and metric embeddings (Johnson & Lindenstrauss, 1984) to measure geometric distortion (Tenenbaum et al., 2000) can be extended to graph-structured data to measure topological distortion. A distortion measure quatifies how the distances between the embedded points deviate from the original distances in the sample space. The deviation can be measured in the form of differences, ratios, or differences in rankings. Different distortion measures differ in how well they preserve the original graph's distances.

Despite extensive efforts to preserve topological properties in graph embedding, most research has concentrated on developing graph embedding algorithms or encoders. At the same time, progress has been made in developing distortion measures to reduce distortion in Euclidean embeddings. Some methods have examined hyperbolic embeddings for hierarchical and tree-like structures (Nickel & Kiela, 2017). However, a universal distortion measure that is independent of embedding algorithms and sufficiently versatile for various learning tasks has yet to be established.

The goal of this paper is to show that minimum-distortion is a key principle for graph embedding. The work does not involve proposing a new embedding algorithm but focuses on adding regularization on distortion to existing algorithms. Minimum-distortion graph embedding inherently prevents embedding collapse, resulting in dispersed node representations. Embeddings with distortion regularization outperform their counterparts in downstream tasks, as demonstrated by experimental results, by effectively preserving both local and global topological properties of the input graph rather than prioritizing representation diversity.

The primary contributions of this paper are outlined below, with the first point serving as the primary motivation:

1. Demonstrate that minimum-distortion is a fundamental principle for graph embedding, and introduce a generic distortion measure that effectively preserves distance relationships during embedding.

2. Assess different distortion measures and their relative strengths.

3. Established the equivalence between Gaussian kernel embedding and a particular form of minimum-distortion embedding.

## 2 PROXIMITY-PRESERVING GRAPH EMBEDDING

Proximity-preserving is a widely used strategy in graph embedding. In such embedding methods, proximity in a graph is preserved when embedded into a manifold in an embedding space. Mathematically, proximity-preserving graph embedding can be described as a mapping $f : \mathbb{V} \to \mathbb{R}^d$, where $\mathbb{V}$ is the set of nodes and $\mathbb{R}^d$ is the $d$-dimensional embedding space, such that a proximity function (e.g., adjacency, cosine similarity of node feature vectors, or path-based similarity) in the graph is approximated by a proximity function in the embedding space. For example, first-order proximity aims to minimize $\|f(u) - f(v)\|_2$ (or maximize cosine similarity) if nodes $u$ and $v$ are connected in the graph, while second-order proximity aims to minimize $\|f(u) - f(v)\|_2$ (or maximize cosine similarity) if $u$ and $v$ share many neighbors. Large-scale Information Network Embedding (LINE) in Tang et al. (2015) is an example of preserving both first-order and second-order proximity.

Although proximity-preserving graph embedding has a clear conceptual definition, there's no universal implementable definition for proximity. Choosing which type of proximity to prioritize is subjective and task-dependent. In this paper, we leave the choice of proximity definition to the application end, and discuss another aspect of proximity-preserving graph embedding—distortion. As Pei et al. (2020) pointed out, while graph topological patterns are preserved, geometry patterns may be distorted. In our discussion, we use the concept of distance — the opposite of proximity—to discuss how distortion can be prevented. While previous work Pei et al. (2020) uses curvature regulation to minimize the distortion indirectly, we aim to directly minimize distortion during the training of the algorithm.

Ideally, an embedding algorithm $f$ should strictly preserve pairwise distances after removing scaling effects. A less restrictive requirement is that $f$ should at least preserve the rankings of pairwise

distances. The *mean average precision* (MAP) is a local measure to quantify deviations in rankings, and the Spearman's footrule distance, or the F-distance (see section 5 for the formal definition), is a global measure. However, the orderings of distances cannot be conveniently used in machine learning training due to the difficulty in gradient propagation.

# 3 MINIMUM-DISTORTION GRAPH EMBEDDING

## 3.1 A GENERIC DISTORTION MEASURE

A continuously differentiable function that quantifies the distortion is necessary to enable direct distortion regularization. Let $f : \mathbb{V} \to \mathbb{R}^d$ represent an embedding algorithm. We omit specific choices of distance measures, using symbolic notation $d_H$ and $d_S$ to denote the embedding space distance and the sample space distance, respectively. Let $\rho(i, j)$ represent the normalized ratio of the distance in the embedding space to the distance in the sample space, defined as:

$$\rho(i, j) = \frac{d_H(f(i), f(j))}{d_S(i, j)} \frac{\sum\limits_{u,v \in \mathbb{V}} d_S(u, v)}{\sum\limits_{u,v \in \mathbb{V}} d_H(f(u), f(v))}, \text{ for } i \neq j. \tag{1}$$

Using the ratio of normalized distances has the benefit that, under uniform scaling of the distances, the ratio $\rho(i, j) = 1$.

The network-wide distortion is defined as:

$$\mathcal{D}_\rho(f) = \frac{1}{\binom{n}{2}} \sum_{i \neq j} |\rho(i, j) - 1| \tag{2}$$

If all embeddings collapse to one point at some iteration during training, $\sum\limits_{u,v \in \mathbb{V}} d_H(f(u), f(v)) = 0$. To avoid dividing by zero, we add a small constant $\epsilon$ to it, then $\frac{d_H(f(i), f(j))}{\sum\limits_{u,v \in \mathbb{V}} d_H(f(u), f(v)) + \epsilon} = \frac{0}{\epsilon} = 0$, thus creating significant distortion. Minimizing distortion will drive $\rho(i, j)$ close to 1 and move the embeddings away from the collapsing point. Therefore, minimizing distortion implicitly scatters the embeddings.

A distortion measure should be invariant under uniform scaling of distances in either the source or target space. This is formally defined as the scale-invariance property.

**Definition 1** (Scale-Invariance). For an embedding algorithm $f : (\mathbb{V}, d_S) \to (\mathbb{R}^d, d_H)$, the distortion measure $\mathcal{D}(f, d_S, d_H)$ is scale-invariant if $\mathcal{D}(f, \lambda d_S, \mu d_H) = \mathcal{D}(f, d_S, d_H)$, for all $\lambda, \mu > 0$.

Being scale-invariant means the distortion measure depends only on the relative relationships between distances, not on their absolute scales; therefore, embeddings that differ solely in units of measurement will have zero distortion according to $\mathcal{D}$.

**Theorem 1.** $\mathcal{D}_\rho$ *is scale-invariant.*

Proof for Theorem 1 can be found in Appendix A.

## 3.2 LEARNING MINIMUM-DISTORTION EMBEDDING

The distortion loss corresponding to the distortion measure $\mathcal{D}_\rho$ is given as:

$$\mathcal{L}_\rho = \frac{1}{\binom{n}{2}} \sum_{i \neq j} (\rho(i, j) - 1)^2 \tag{3}$$

To learn the minimum-distortion embeddings, we can directly minimize the distortion loss during training. $\mathcal{L}_\rho$ can be used either as the sole optimization criterion or as a regularization term included in the total loss function.

The choises of distance measures $d_H$ and $d_S$ are task-dependent. Sample space distance can be the distance between two nodes on the graph. A widely used distance measure is the shortest path distance. It can also be the distance between two feature vectors, e.g., the Eucliden distance or cosine dissimilarity between two feature vectors. Embedding space distance is the distance between two embeddings. It can also be the Euclidean distance or cosine dissimilarity.

The cosine dissimilarity used in this paper is defined as:

$$d(v_1, v_2) = 1 - \frac{\langle v_1, v_2 \rangle}{\|v_1\|\|v_2\|}$$

This definition satisfies the properties of non-negativity (i.e., $d(v_1, v_2) \geq 0$), symmetry (i.e., $d(v_1, v_2) = d(v_2, v_1)$), and identity (i.e., $d(v, v) = 0$).

## 4 GAUSSIAN KERNEL EMBEDDING IS A TYPE OF MINIMUM-DISTORTION EMBEDDING

Gaussian kernels are frequently used in machine learning and statistics to represent data points or distributions in a high-dimensional feature space (Sriperumbudur et al., 2010; Gretton et al., 2012; Muandet et al., 2017; Zhuang et al., 2023; Li & Yuan, 2024; Zhao et al., 2025). In this paper, we demonstrate that Gaussian kernel embedding is a type of minimum-distortion embedding.

### 4.1 GAUSSIAN KERNEL EMBEDDING (GKE)

Let $F$ and $d$ denote the feature dimensions in the sample space and embedding space, respectively. Gaussian kernel embedding uses the following embedding algorithm to map a vector $x \in \mathbb{R}^F$ to its embedding vector $h \in \mathbb{R}^d$,

$$h_i(x) = c_i \exp\left(-\frac{\|x - \mu_i\|^2}{2\sigma^2}\right), \quad \text{for } i = 1 \dots d. \tag{4}$$

If we normalize $h$ such that $\|h\|_2 = 1$, choose $d_H(h(x), h(y)) = 1 - \langle h(x), h(y) \rangle$, and $d_S(x, y) = \|x - y\|_2$, let $z = \frac{x+y}{2}$, then we have the following relationship between $d_S$ and $d_H$:

$$1 - d_H(h(x), h(y)) = \langle h(x), h(y) \rangle = \left(\sum_{i=1}^{d} c_i^2 \exp\left(-\frac{\|z - \mu_i\|^2}{\sigma^2}\right)\right) \cdot \exp\left(-\frac{\|x - y\|^2}{2\sigma^2}\right). \tag{5}$$

Let $S_d$ denote the first term:

$$S_d = \sum_{i=1}^{d} c_i^2 \exp\left(-\frac{\|z - \mu_i\|^2}{\sigma^2}\right) \tag{6}$$

By setting $c_i^2 = \frac{1}{d}$, $S_d$ becomes the normalized sum of Gaussian PDFs,

$$S_d = \frac{1}{d} \sum_{i=1}^{d} \exp\left(-\frac{\|z - \mu_i\|^2}{\sigma^2}\right) \tag{7}$$

If the points $\mu_i$ are i.i.d. samples from a probability distribution with density $p(\mu)$, $S_d$ is the sample mean of the random variable $\exp\left(-\frac{\|z - \mu_i\|^2}{\sigma^2}\right)$. We next analyze the convergence property of $S_d$.

### 4.1.1 CASE 1: $d \to \infty$

By the law of large numbers, the sample mean $S_d$ converges almost surely to the expectation as $d \to \infty$ provided that the expectation is finite:

$$S_d \to \mathbb{E}_{\mu \sim p}\left[\exp\left(-\frac{\|z-\mu\|^2}{\sigma^2}\right)\right] = \int_{\mathbb{R}^F} p(\mu)\exp\left(-\frac{\|z-\mu\|^2}{\sigma^2}\right) d\mu. \tag{8}$$

Let $C = \mathbb{E}_{\mu \sim p}\left[\exp\left(-\frac{\|z-\mu\|^2}{\sigma^2}\right)\right]$ denote the expectation. We show that $C$ is bounded: Since Gaussian kernel is bounded ( i.e., $\exp\left(-\frac{\|z-\mu\|^2}{\sigma^2}\right) \le 1$) and integrable,

$$\int_{\mathbb{R}^F} \exp\left(-\frac{\|z-\mu\|^2}{\sigma^2}\right) d\mu = \left(\pi\sigma^2\right)^{F/2}, \tag{9}$$

and $p(\mu)$ satisfies $\int_{\mathbb{R}^F} p(\mu)d\mu = 1$, the integral of the expectation is bounded:

$$0 \le C \le \int_{\mathbb{R}^F} p(\mu) \cdot 1 \, d\mu = 1 \tag{10}$$

The exponential decay of the kernel ensures $C$ is finite for any valid probability density $p(\mu)$.

Therefore, with $d \to \infty$,

$$d_H(h(x), h(y)) \to 1 - C\exp\left(-\frac{\|x-y\|^2}{2\sigma^2}\right). \tag{11}$$

$d_H$ satisfies the following properties:

- Non-negativity: $d_H(h(x), h(y)) \ge 0$.
- Symmetry: $d_H(h(x), h(y)) = d_H(h(y), h(x))$.
- If we set $c_i^2 = \frac{1}{Cd}$, then

$$d_H(h(x), h(y)) \to 1 - \exp\left(-\frac{\|x-y\|^2}{2\sigma^2}\right), \tag{12}$$

  which leads to additional properties:
    - Identity: $d_H(h(x), h(x)) = 0$.
    - Identity of indiscernibles: $d_H(h(x), h(y)) = 0 \iff x = y$.

### 4.1.2 CASE 2: FINITE $d$

For finite dimensional embedding, $S_d$ as the sample mean of $d$ i.i.d. random variables has the following expectation and variance:

$$\mathbb{E}[S_d] = \mathbb{E}_{\mu \sim p}\left[\exp\left(-\frac{\|z-\mu\|^2}{\sigma^2}\right)\right] = C, \quad \text{Var}(S_d) = \frac{1}{d}\text{Var}\left(\exp\left(-\frac{\|z-\mu\|^2}{\sigma^2}\right)\right). \tag{13}$$

Since $\exp\left(-\frac{\|z-\mu\|^2}{\sigma^2}\right) \le 1$, the variance is bounded, and $\text{Var}(S_d) \propto \frac{1}{d}$. Thus, $S_d$ concentrates around $C$ as $d$ increases.

### 4.2 LEARNING GAUSSIAN KERNEL EMBEDDING

To learn the Gaussian kernel embedding, we use the following loss function and use $\sigma^2$ as a hyperparameter:

$$\mathcal{L}_{\text{GKE}} = \sum_{x \ne y}\left(\langle h(x), h(y)\rangle - \exp\left(-\frac{\|x-y\|^2}{2\sigma^2}\right)\right)^2. \tag{14}$$

This loss function corresponds to a distortion measure $\mathcal{D}_{\text{GKE}}(h)$:

$$\mathcal{D}_{\text{GKE}}(h) = \sum_{x \neq y} \left| \langle h(x), h(y) \rangle - \exp\left(-\frac{\|x-y\|^2}{2\sigma^2}\right) \right|$$

$$= \sum_{x \neq y} \left| d_H(h(x), h(y)) - \left(1 - \exp\left(-\frac{d_S(x,y)^2}{2\sigma^2}\right)\right) \right| \tag{15}$$

with the sample space employing Euclidean distance and the embedding space employing cosine dissimilarity as distance measures. Therefore, Gaussian kernel embedding with loss function $\mathcal{L}_{\text{GKE}}$ is a type of minimum-distortion embedding with distortion function $\mathcal{D}_{\text{GKE}}$.

**Theorem 2** (Large dimension embedding). *As $d \to \infty$, $\mathcal{L}_{GKE} \to 0$ and $\mathcal{D}_{GKE} \to 0$, Gaussian Kernel Embedding preserves the orderings of pairwise distances via a nonlinear mapping.*

## 5   COMPARISON OF DISTORTION MEASURES

### 5.1   CONDITIONS FOR OPTIMALITY OF DISTORTION MEASURES

We analyze whether an embedding $f$ with zero distortion under a specific distortion measure ensures strict distance preserving. We discuss four representative works, each representing a category.

- Distortion measure $\mathcal{D}_\rho(f)$ defined in Equation (2).
  $\mathcal{D}_\rho = 0$ requires $\rho(i,j) = 1$, which requires

  $$d_H(f(i), f(j)) = k \cdot d_S(i,j), \quad \text{with } k = \frac{\sum_{u \neq v} d_H(f(u), f(v))}{\sum_{u \neq v} d_S(u,v)}.$$

  This requires exact linear scaling of all pairwise distances, preserving both distances and their orderings.

- GKE distortion measure $\mathcal{D}_{\text{GKE}}$ defined in Equation (15).
  $\mathcal{D}_{\text{GKE}} = 0$ requires

  $$d_H(h(x), h(y)) = 1 - \exp\left(-\frac{d_S(x,y)^2}{2\sigma^2}\right).$$

  $d_H$ increases in $d_S$, preserving the orderings of distances exactly, but preserving the magnitudes of distances via a nonlinear mapping.

- Spearman's footrule as a distortion measure (Diaconis & Graham, 1977; Spearman, 1906).

  $$\mathcal{D}_F(f) = \frac{2}{n(n-1)} \sum_{i=1}^{n-1} F(\pi_S^i, \pi_H^i) = \frac{2}{n(n-1)} \sum_{i=1}^{n-1} \sum_{j=i+1}^{n} \left| \pi_S^i(j) - \pi_H^i(j) \right|,$$

  where $\pi_S^i$ and $\pi_H^i$ are permutations of nodes by $d_S(i,:)$ and $d_H(f(i),:)$, respectively, and $\pi(j)$ is the rank of item $j$ in permutation $\pi$.
  $\mathcal{D}_F = 0$ requires identical orderings in $\pi_S^i$ and $\pi_H^i$, preserving the orderings of distances but not necessarily the magnitudes of distances.

- Mean Average Precision (MAP), proposed as a distortion measure in Nickel & Kiela (2017).

  $$\text{MAP}(f) = \frac{1}{|\mathbb{V}|} \sum_{v \in \mathbb{V}} \frac{1}{|N_v|} \sum_{i=1}^{|N_v|} \frac{|N_v \cap H_v(i)|}{|H_v(i)|}.$$

  $N_v$ is the set of neighbors of node $v$ on the graph, and $|N_v|$ is the degree of node $v$; $H_v(i)$ is the smallest set of nodes that includes the $i$-th nearest neighbor of $v$ in the embedding space.
  Zero distortion means MAP $= 1$, which requires the $|N_v|$ closest nodes in the embedding space to be identical to $N_v$, focusing only on local neighborhood preservation.

## 5.2 STRICTNESS OF DISTORTION MEASURES

A distortion measure $\mathcal{D}$ quantifies the deviation between the distances $d_S(x, y)$ in the original space and the distances $d_H(f(x), f(y))$ in the target space. Next, we analyze how sensitive a distortion measure is to various deviations.

**Definition 2** (Relative Strictness of Distortion Measures: $\mathcal{A} \succ \mathcal{B}$). A distortion measure $\mathcal{A}$ is stricter than a distortion measure $\mathcal{B}$, denoted as $\mathcal{A} \succ \mathcal{B}$, if optimality in $\mathcal{A}$ implies optimality in $\mathcal{B}$, but the converse is not true.

The condition *optimality in $\mathcal{A}$ implies optimality in $\mathcal{B}$, but the converse is not true* can be expressed using set notation: define $S_{\mathcal{A}} = \{f \mid \mathcal{A}(f) = 0\}$ and $S_{\mathcal{B}} = \{f \mid \mathcal{B}(f) = 0\}$, then $S_A \subset S_B$. $\mathcal{A}$ is stricter because its optimality condition is more restrictive; therefore, the set $S_{\mathcal{A}}$ is smaller.

**Theorem 3.** $\mathcal{D}_\rho \succ \mathcal{D}_F \succ MAP$, and $\mathcal{D}_{GKE} \succ \mathcal{D}_F \succ MAP$.

**Definition 3** (Strictness Measure for Deviation from Linear Maps: $\mathcal{A} \overset{L}{\succ} \mathcal{B}$). If the optimality of $\mathcal{A}$ requires $f$ to be a linear map that scales distances by a constant $k > 0$, while the optimality of $\mathcal{B}$ allows functions that are not scaled maps, then $\mathcal{A}$ is a stricter measure for deviation from linear maps, denoted as $\mathcal{A} \overset{L}{\succ} \mathcal{B}$.

$\mathcal{A} \overset{L}{\succ} \mathcal{B}$ indicates that $\mathcal{A}$ imposes a more stringent condition for zero distortion and is more sensitive to deviation from a linear map, assigning non-zero distortion to any function that fails to maintain a constant distance ratio, while $\mathcal{B}$ may tolerate such deviation.

**Theorem 4.** $\mathcal{D}_\rho \overset{L}{\succ} \mathcal{D}_{GKE}$, $\mathcal{D}_\rho \overset{L}{\succ} \mathcal{D}_F$, and $\mathcal{D}_\rho \overset{L}{\succ} MAP$.

Proofs for theorems are available in Appendix A.

## 6 EXPERIMENT

We evaluate our model on three node-level tasks — node classification, node clustering, and node similarity search — following the evaluation protocol of Lee et al. (2022). We also evaluate it on one link-level task — link prediction — following the evaluation protocol of Kipf & Welling (2016).

### 6.1 NODE CLASSIFICATION

Distortion regularization is used in both supervised and unsupervised settings for node classification.

#### 6.1.1 UNSUPERVISED EMBEDDING GENERATION AND SUPERVISED CLASSIFICATION

In the unsupervised setting, we pretrain an encoder model to generate embeddings and then use cross-entropy loss $\mathcal{L}_{\text{ce}}$ on labeled data for node classification. There is no backpropagation of gradients to the pretrained model from the second stage.

For the encoder, we adopt the SGRL-style online-target model, but add a distortion regularization term in both the online and target models, where the distance measures for both $d_S$ and $d_H$ are cosine dissimilarity, $\mathcal{L}_{\text{SGRL}}^{\text{online}}$ and $\mathcal{L}_{\text{SGRL}}^{\text{target}}$ are the original loss functions used in SGRL:

$$\mathcal{L}_{\text{total}}^{\text{online}} = \mathcal{L}_{\text{SGRL}}^{\text{online}} + \lambda_{\text{online}} \mathcal{L}_\rho, \quad \mathcal{L}_{\text{total}}^{\text{target}} = \mathcal{L}_{\text{SGRL}}^{\text{target}} + \lambda_{\text{target}} \mathcal{L}_\rho,$$

The decoder model is a simple logistic regression classifier, which is the same as in He et al. (2024a).

We label our method as Ours-SGRL and compare the original SGRL with Ours-SGRL on five benchmark datasets, including WikiCS (Mernyei & Wiki-CS, 2020), Amazon Computers and Amazon Photo (McAuley et al., 2015), Coauthor-CS and Coauthor-Physics (Sinha et al., 2015). Data splitting follows the same protocol as in He et al. (2024a); Lee et al. (2022); Thakoor et al. (2021).

Table 1 shows the averages and standard deviations of the F1-scores for baselines before and after distortion regularization. Results are based on training 200 epochs for each. It demonstrates that adding distortion regularization improves the performance of the original model for the most part. Additional comparisons with more baselines are shown in Table 4 (See Appendix B.1).

**Table 1:** Node classification accuracy. X, A, Y denote the node attributes, adjacency matrix, and labels in the datasets. The '+' notation is used in two-stage training methods — unsupervised embedding generation followed by supervised classification. (X, A) + Y denotes X and A are used to generate node embeddings through an unsupervised approach, and these embeddings are then used with labeled data Y to train a classifier in a supervised manner. The best results in each category are highlighted in bold. Results for one-stage training baselines are from He et al. (2024a).

| Method | Data | WikiCS | Computers | Photo | Co.CS | Co.Physics |
|--------|------|--------|-----------|-------|-------|------------|
| | | | **Two-stage training** | | | |
| SGRL | (X, A) + Y | $79.45 \pm 0.10$ | $90.19 \pm 0.11$ | $93.11 \pm 0.06$ | $\mathbf{93.45} \pm 0.03$ | $\mathbf{96.01} \pm 0.04$ |
| Ours-SGRL | (X, A) + Y | $\mathbf{79.47} \pm 0.10$ | $\mathbf{90.23} \pm 0.08$ | $\mathbf{93.32} \pm 0.10$ | $\mathbf{93.45} \pm 0.05$ | $95.99 \pm 0.04$ |
| | | | **One-stage training** | | | |
| GCN | X, A, Y | $77.19 \pm 0.12$ | $86.51 \pm 0.54$ | $92.42 \pm 0.22$ | $93.03 \pm 0.31$ | $95.65 \pm 0.16$ |
| Ours-GCN | X, A, Y | $\mathbf{78.71} \pm 0.47$ | $\mathbf{89.29} \pm 0.52$ | $\mathbf{92.95} \pm 0.48$ | $\mathbf{93.07} \pm 0.19$ | $\mathbf{95.87} \pm 0.09$ |

### 6.1.2 SEMI-SUPERVISED NODE CLASSIFICATION

In semi-supervised node classification, we use an end-to-end training strategy: use the cross-entropy loss $\mathcal{L}_{\text{ce}}$ as the primary loss, and incorporate the distortion loss $\mathcal{L}_\rho$ as a regularization term with a hyperparameter $\lambda_1$ controlling the strength of regularization. Assume there are $C$ classes and the training set includes $m$ data points. The model is trained with the following loss function.

$$\mathcal{L}_{\text{total}}^{\text{node}} = \mathcal{L}_{\text{ce}}^{\text{node}} + \lambda_1 \mathcal{L}_\rho \quad = -\frac{1}{m} \sum_{i=1}^{m} \sum_{k=1}^{C} y_{i,k} \log(\hat{p}_{i,k}) + \lambda_1 \mathcal{L}_\rho \tag{16}$$

We adopt the GCN model for end-to-end training and add the distortion loss $\mathcal{L}_\rho$ as a regularization term. Our method is labeled as Ours-GCN in Table 1. Data splitting and result evaluation follow the same protocols as in 6.1.1. Compared to the baseline GCN (Kipf & Welling, 2017), Ours-GCN shows improved classification performance across all datasets.

### 6.2 CLUSTERING AND SIMILARITY SEARCH

Clustering and similarity search are two node-level tasks used to evaluate unsupervised learning. We follow the protocol of Lee et al. (2022) for evaluating results.

**Table 2:** *(left)* Performance on clustering measured by NMI and h-score. *(Right)* Performance on similarity search measured by Sim@5 and Sim@10. Optimal results are shown in bold. Results are from running 200 training epochs.

| | | SGRL | Ours-SGRL | | SGRL | Ours-SGRL |
|--------|---------|--------|-----------|--------|--------|-----------|
| WikiCS | NMI | 0.4239 | **0.4241** | Sim@5 | **0.7968** | 0.7967 |
| | h-score | 0.4426 | **0.4430** | Sim@10 | **0.7825** | 0.7825 |
| Computers | NMI | **0.5282** | 0.5273 | Sim@5 | 0.8883 | **0.8900** |
| | h-score | **0.5637** | 0.5553 | Sim@10 | 0.8785 | **0.8809** |
| Photo | NMI | 0.6719 | **0.6730** | Sim@5 | 0.9186 | **0.9200** |
| | h-score | 0.6736 | **0.6742** | Sim@10 | 0.9115 | **0.9136** |
| Co.CS | NMI | 0.7512 | **0.7540** | Sim@5 | 0.9036 | **0.9071** |
| | h-score | 0.7837 | **0.7854** | Sim@10 | 0.8971 | **0.9020** |
| Co.Physics | NMI | 0.7186 | **0.7288** | Sim@5 | **0.9523** | 0.9518 |
| | h-score | 0.7324 | **0.7418** | Sim@10 | **0.9480** | 0.9474 |

To evaluate clustering performance, we use two metrics: Normalized Mutual Information (NMI) and Homogeneity score (h-score). NMI assesses the mutual information between the true labels and the predicted cluster assignments, scaled to fall between 0 and 1. A higher NMI signifies better clustering, with 1 indicating perfect agreement and 0 indicating no mutual information. In contrast, h-score measures how much each cluster contains members from a single class based on true class labels. It ranges from 0 to 1, where 1 signifies perfectly homogeneous clusters (all members in a cluster belong to the same class) and 0 indicates poor homogeneity. The left panel of Table 2 shows the comparison between SGRL and Ours-SGRL. Additional comparisons with more baselines are

shown in Table 5 (See Appendix B.2). To evaluate the performance of node similarity search, we use Sim@5 and Sim@10. Sim@n measures the proportion of the top $n$ nearest neighbors based on cosine similarity that share the same true label as the query node, averaged over multiple queries. The right panel of Table 2 shows that Ours-SGRL generally outperforms SGRL. Comparisons with more baselines are shown in Table 6 in Appendix B.3.

## 6.3 LINK PREDICTION

We adopt the GAE model in Kipf & Welling (2016) for link prediction, and add distortion regularization into the loss function. The encoder consists of two graph convolutional layers. The decoder first calculates the inner product between two normalized node vectors ($h_i \in \mathbb{R}^d$ is the normalized row vector of $\boldsymbol{H}$), and then uses the logistic sigmoid function as the activation function:

$$\text{Encoder: } \boldsymbol{H} = GCN(\boldsymbol{X}, \boldsymbol{A}), \ \boldsymbol{H} \in \mathbb{R}^{n \times d}, \quad \text{Decoder: } p(A_{ij} = 1 | h_i, h_j) = \sigma(h_i^\top h_j).$$

Let $\tilde{E}$ denote the set of links in the training set. The link prediction model is trained as follows:

$$\mathcal{L}_{\text{total}}^{\text{link}} = \mathcal{L}_{\text{ce}}^{\text{link}} + \lambda_2 \mathcal{L}_{\text{dis}} \quad = -\frac{1}{|\tilde{E}|} \sum_{(i,j) \in \tilde{E}} A_{ij} \log p(A_{ij} | h_i, h_j) + \lambda_2 \mathcal{L}_{\text{dis}}. \tag{17}$$

The first term is the cross-entropy loss for link prediction; the second term is the distortion loss $\mathcal{L}_{\text{dis}}$, representing either $\mathcal{L}_\rho$ or $\mathcal{L}_{\text{GKE}}$.

To evaluate link prediction performance, we examine the area under the ROC curve (AUC) and average precision (AP) over three datasets: Cora, CiteSeer, and PubMed (Sen et al., 2008). Models are trained on an incomplete version of these datasets where parts of the citation links have been removed, while all node features are kept. Link splitting follows the protocol in Kipf & Welling (2016): the training set contains 85% of the citation links; the validation and test sets are from previously removed edges and the same number of randomly sampled pairs of disconnected nodes (non-edges). The validation and test sets contain 5% and 10% of citation links, respectively.

We compare the link prediction results with the GAE and VGAE models in Kipf & Welling (2016), with a grid search for hyperparameters: hidden dimension $d \in \{16, 32, 64\}$, learning rate $lr \in \{0.01, 0.02\}$, and regularization term $\lambda_2 \in \{0.25, 0.5, 0.75, 1.0, 2.0\}$. Weight initialization follows the method in Glorot & Bengio (2010). We train these models for 200 epochs with Adam optimizer. Our methods are labeled as Ours-$\mathcal{L}_\rho$ and Ours-$\mathcal{L}_{\text{GKE}}$ in Table 3. In Ours-$\mathcal{L}_\rho$, $\mathcal{L}_\rho$ is used to substitute $\mathcal{L}_{\text{dis}}$ in Eq. (17), and cosine dissimilarity is used as the distance measures for both $d_S$ and $d_H$ in $\mathcal{L}_\rho$. In Ours-$\mathcal{L}_{\text{GKE}}$, $\mathcal{L}_{\text{GKE}}$ is used. Across three datasets, our methods consistently outperforms the baseline GAE, with Ours-$\mathcal{L}_\rho$ performing the best and Ours-$\mathcal{L}_{\text{GKE}}$ the second best.

**Table 3:** Link prediction performance (AUC and AP, in percentage) on Cora, CiteSeer, and PubMed datasets. Optimal results are shown in bold and suboptimal results are in italics. Results are from training 200 epochs.

| | Cora | | CiteSeer | | PubMed | |
|---|---|---|---|---|---|---|
| | AUC | AP | AUC | AP | AUC | AP |
| GAE | $97.59 \pm 0.67$ | $97.03 \pm 0.94$ | $96.66 \pm 0.38$ | $96.02 \pm 0.50$ | $97.34 \pm 0.26$ | $96.95 \pm 0.26$ |
| VGAE | $92.62 \pm 1.83$ | $91.98 \pm 1.78$ | $89.71 \pm 1.42$ | $88.87 \pm 1.67$ | $93.60 \pm 0.60$ | $93.23 \pm 0.57$ |
| Ours-$\mathcal{L}_{\text{GKE}}$ | *$97.79 \pm 0.23$* | *$97.31 \pm 0.33$* | *$96.99 \pm 0.36$* | *$96.49 \pm 0.48$* | *$97.82 \pm 0.16$* | *$97.46 \pm 0.15$* |
| Ours-$\mathcal{L}_\rho$ | **$98.52 \pm 0.38$** | **$98.17 \pm 0.48$** | **$98.76 \pm 0.19$** | **$98.48 \pm 0.33$** | **$98.73 \pm 0.09$** | **$98.46 \pm 0.15$** |

## 7 CONCLUSION

In this paper, we introduce a distortion regularization method for graph embedding. This additive approach directly enhances the performance of many SOTA methods, with consistent improvements observed across several downstream tasks, indicating a negative correlation between distortion in embeddings and performance. We also provide an explanation for the strong performance of Gaussian kernel embedding through the perspective of minimum distortion. Future work will investigate the relationship between distortion and performance across a wider range of embedding methods.

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

APPENDIX

# A    PROOFS FOR THEOREMS

## A.1    PROOF FOR THEOREM 1

Let $d'_S(x,y) = \lambda d_S(x,y)$ for all pairs $(x,y)$. Then,

$$\frac{\sum\limits_{u,v} d'_S(u,v)}{d'_S(i,j)} = \frac{\sum\limits_{u,v} d_S(u,v)}{d_S(i,j)}.$$

Similarly, let $d'_H(x,y) = \mu d_H(x,y)$ for all pairs $(x,y)$. Then,

$$\frac{d'_H(f(i),f(j))}{\sum\limits_{u,v} d'_H(f(u),f(v))} = \frac{d_H(f(i),f(j))}{\sum\limits_{u,v} d_H(f(u),f(v))}.$$

In either case, the $\rho(i,j)$ defined in Eq. (1) remains the same. Therefore, $\mathcal{D}_\rho(f, \lambda d_S, \mu d_H) = \mathcal{D}_\rho(f, d_S, d_H)$. This proves that $\mathcal{D}_\rho$ is scale-invariant.

## A.2    PROOF FOR THEOREM 2

When $d \to \infty$, from Eq. (12), we have

$$d_H(h(x),h(y)) \to 1 - \exp\left(-\frac{\|x-y\|^2}{2\sigma^2}\right),$$

$$\langle h(x),h(y)\rangle \to \exp\left(-\frac{\|x-y\|^2}{2\sigma^2}\right),$$

therefore, $\mathcal{L}_{\text{GKE}} \to 0$ and $\mathcal{D}_{\text{GKE}} \to 0$.

Let $d_S(x,y) = \|x-y\|^2$. When $\mathcal{D}_{\text{GKE}} = 0$,

$$d_H(h(x),h(y)) \to 1 - \exp\left(-\frac{d_S(x,y)^2}{2\sigma^2}\right),$$

therefore, for any two pairs $(x,y)$ and $(u,v)$, $d_H(h(x),h(y)) \le d_H(h(u),h(v))$ if and only if $d_S(x,y) \le d_S(u,v)$, thus, the orderings of pairwise distances are preserved.

## A.3    PROOF FOR THEOREM 3

**1).**    Show that $\mathcal{D}_\rho \succ \mathcal{D}_F$: The optimal value for $\mathcal{D}_\rho$ is $\mathcal{D}_\rho = 0$. If $\mathcal{D}_\rho(f) = 0$ for an embedding algorithm $f$, then it makes $\rho(i,j) = 1$ for all pairs $i \ne j$. Therefore, there exists a constant $k > 0$ such that $d_H(f(i),f(j)) = k \cdot d_S(i,j)$. Since uniform scaling preserves the order of the distances in the sample space, $\mathcal{D}_F(f) = 0$, achieving optimality in $\mathcal{D}_F$.

**2).**    Show that $\mathcal{D}_{\text{GKE}} \succ \mathcal{D}_F$: The optimal value for $\mathcal{D}_{\text{GKE}}$ is $\mathcal{D}_{\text{GKE}} = 0$. If $\mathcal{D}_{\text{GKE}}(f) = 0$ for an embedding algorithm $f$, then $d_H(h(x),h(y)) = 1 - \exp\left(-\frac{d_S(x,y)^2}{2\sigma^2}\right)$. Since $d_H$ increases monotonicly with $d_S$, $f$ preserves the order of distances in the sample space, therefore $\mathcal{D}_F(f) = 0$, achieving optimality in $\mathcal{D}_F$.

**3).**    Show that $\mathcal{D}_F \succ \text{MAP}$: The optimal value for $\mathcal{D}_F$ is $\mathcal{D}_F = 0$, and the optimal value for MAP is MAP $= 1$. $\mathcal{D}_F = 0$ requires the distance orderings by $d_H$ to completely agree with the orderings by $d_S$. If $\mathcal{D}_F(f) = 0$ for an embedding algorithm $f$, then for each node $v$, $H_v(i) \subseteq N_v$ for $i = 1, \ldots, |N_v|$, provided that the distance measure $d_S$ used in computing $\mathcal{D}_F$ is consistent with the distance measure used in defining $N_v$ on the graph. Therefore, MAP $(f) = 1$, achieving optimality in MAP.

## A.4 PROOF FOR THEOREM 4

$\mathcal{D}_\rho(f) = 0$ requires $f$ to be a linear map that scales distances by a constant $k > 0$: $d_H(f(i), f(j)) = k \cdot d_S(i,j)$. Any deviations from a linear map will result in $\mathcal{D}_\rho(f) > 0$.

$\mathcal{D}_{\text{GKE}}(f) = 0$ allows $f$ to be a nonlinear map: $d_H = 1 - \exp\left(-\frac{d_s^2}{2\sigma^2}\right)$ makes $\mathcal{D}_{\text{GKE}} = 0$, but there does not exist a constant $k > 0$ such that $d_H = kd_S$ for all pairs.

$\mathcal{D}_F(f) = 0$ allows $f$ to be non-linear map: there exists a non-linear map $f$ such that $\pi_S^i(j) = \pi_H^i(j)$, but $d_H(f(i), f(j)) \neq kd_S(i,j)$.

Since there exists a non-linear map $f$ that achieves $\mathcal{D}_F(f) = 0$, such $f$ will also achieve $\text{MAP} = 1$.

## B PERFORMANCE COMPARISON WITH MORE BASELINE METHODS

### B.1 NODE CLASSIFICATION

Additional comparisons with more baselines are shown in Table 4. SGRL-$n$ stands for training SGRL for $n$ epochs. Ours-SGRL and SGRL-200 are the results of 200 epochs of training.

Results for SGRL-1000 and other baseline methods are from He et al. (2024a), which shows SGRL-1000 is the best among all other baselines including Node2vec (Grover & Leskovec, 2016), Deepwalk (Perozzi et al., 2014), DGI (Velickovic et al., 2019), GRACE (Zhu et al., 2020), GCA (Zhu et al., 2021), iGCL (Li et al., 2023), GBT (Bielak et al., 2022), BGRL (Thakoor et al., 2021), AFGRL (Lee et al., 2022), and MVGRL (Hassani & Ahmadi, 2020).

The WikiCS dataset includes 20 predefined training-validation-testing splits. Classification results using these preset splits are shown for both the baseline methods and our methods. For other datasets, since there are no standard data splits, we use random 10%-90% splits.

Hyperparameters for distortion regularization are chosen through grid search: in Ours-GCN, $\lambda_1 \in \{0.025, 0.05, 0.1, 0.25\}$. In Ours-SGRL, $\lambda_{\text{online}} \in \{0.5, 0.7, 0.9\}$ and $\lambda_{\text{target}} \in \{0.05, 0.1\}$. The hyperparameters of the original SGRL model remain unchanged.

**Table 4:** Node classification accuracy. X, A, Y denote the node attributes, adjacency matrix, and labels in the datasets. The '+' notation is used in two-stage training methods — unsupervised embedding generation followed by supervised classification. (X, A) + Y denotes X and A are used to generate node embeddings through an unsupervised approach, and these embeddings are then used with labeled data Y to train a classifier in a supervised manner.

| Method | Data | WikiCS | Computers | Photo | Co.CS | Co.Physics |
|---|---|---|---|---|---|---|
| | | | Two-stage training | | | |
| Node2vec | A + Y | $71.79 \pm 0.05$ | $84.39 \pm 0.08$ | $89.67 \pm 0.12$ | $85.08 \pm 0.03$ | $91.19 \pm 0.04$ |
| DeepWalk | A + Y | $74.35 \pm 0.06$ | $85.68 \pm 0.06$ | $89.44 \pm 0.11$ | $84.61 \pm 0.22$ | $91.77 \pm 0.15$ |
| GRACE | (X, A) + Y | $77.97 \pm 0.63$ | $86.50 \pm 0.33$ | $92.46 \pm 0.18$ | $92.17 \pm 0.04$ | - |
| DGI | (X, A) + Y | $75.35 \pm 0.14$ | $83.95 \pm 0.47$ | $91.61 \pm 0.22$ | $92.15 \pm 0.63$ | $94.51 \pm 0.52$ |
| BGRL | (X, A) + Y | $76.86 \pm 0.74$ | $89.69 \pm 0.37$ | $93.07 \pm 0.38$ | $92.59 \pm 0.14$ | $95.48 \pm 0.08$ |
| GBT | (X, A) + Y | $76.65 \pm 0.62$ | $88.14 \pm 0.33$ | $92.63 \pm 0.44$ | $92.95 \pm 0.17$ | $95.07 \pm 0.17$ |
| MVGRL | (X, A) + Y | $77.52 \pm 0.08$ | $87.52 \pm 0.11$ | $91.74 \pm 0.07$ | $92.11 \pm 0.12$ | $95.33 \pm 0.03$ |
| GCA | (X, A) + Y | $77.94 \pm 0.67$ | $87.32 \pm 0.50$ | $92.39 \pm 0.33$ | $92.84 \pm 0.15$ | - |
| ProGCL | (X, A) + Y | $78.45 \pm 0.04$ | $89.55 \pm 0.16$ | $93.64 \pm 0.13$ | $93.67 \pm 0.12$ | - |
| AFGRL | (X, A) + Y | $77.62 \pm 0.49$ | $89.88 \pm 0.33$ | $93.22 \pm 0.28$ | $93.27 \pm 0.17$ | $95.69 \pm 0.10$ |
| iGCL | (X, A) + Y | $78.83 \pm 0.08$ | $89.41 \pm 0.06$ | $93.02 \pm 0.06$ | $93.52 \pm 0.04$ | $94.77 \pm 0.20$ |
| SGRL-1000 | (X, A) + Y | $79.40 \pm 0.10$ | $90.23 \pm 0.03$ | $93.95 \pm 0.03$ | $94.15 \pm 0.04$ | $96.23 \pm 0.01$ |
| SGRL-200 | (X, A) + Y | $79.45 \pm 0.10$ | $90.19 \pm 0.11$ | $93.11 \pm 0.06$ | $93.45 \pm 0.03$ | $96.01 \pm 0.04$ |
| Ours-SGRL | (X, A) + Y | $79.47 \pm 0.10$ | $90.23 \pm 0.08$ | $93.32 \pm 0.10$ | $93.45 \pm 0.05$ | $95.99 \pm 0.04$ |
| | | | One-stage training | | | |
| Raw Features | X, Y | $71.98 \pm 0.00$ | $73.81 \pm 0.00$ | $78.53 \pm 0.00$ | $90.37 \pm 0.00$ | $93.58 \pm 0.00$ |
| GCN | X, A, Y | $77.19 \pm 0.12$ | $86.51 \pm 0.54$ | $92.42 \pm 0.22$ | $93.03 \pm 0.31$ | $95.65 \pm 0.16$ |
| Ours-GCN | X, A, Y | $78.71 \pm 0.47$ | $89.29 \pm 0.52$ | $92.95 \pm 0.48$ | $93.07 \pm 0.19$ | $95.87 \pm 0.09$ |

## B.2 CLUSTERING

Results for DGI and SGRL-1000 are from He et al. (2024a), and results for GRACE, BGRL, and AFGRL are from Lee et al. (2022).

**Table 5:** Performance on Clustering in terms of NMI and h-score.

| | | GRACE | DGI | BGRL | AFGRL | SGRL-1000 | SGRL-200 | Ours-SGRL |
|---|---|---|---|---|---|---|---|---|
| WikiCS | NMI | 0.4282 | 0.4312 | 0.3969 | 0.4132 | 0.4188 | 0.4239 | 0.4241 |
| | h-score | 0.4423 | 0.4498 | 0.4156 | 0.4307 | 0.4369 | 0.4426 | 0.4430 |
| Computers | NMI | 0.4793 | 0.4630 | 0.5364 | 0.5520 | 0.5380 | 0.5282 | 0.5273 |
| | h-score | 0.5222 | 0.4836 | 0.5869 | 0.6040 | 0.5705 | 0.5637 | 0.5553 |
| Photo | NMI | 0.6513 | 0.5487 | 0.6841 | 0.6563 | 0.6788 | 0.6719 | 0.6730 |
| | h-score | 0.6657 | 0.5557 | 0.7004 | 0.6743 | 0.6786 | 0.6736 | 0.6742 |
| Co.CS | NMI | 0.7562 | 0.7162 | 0.7732 | 0.7859 | 0.7961 | 0.7512 | 0.7540 |
| | h-score | 0.7909 | 0.7428 | 0.8041 | 0.8161 | 0.8216 | 0.7837 | 0.7854 |
| Co.Physics | NMI | - | 0.6540 | 0.5568 | 0.7289 | 0.7232 | 0.7186 | 0.7288 |
| | h-score | - | 0.6868 | 0.6018 | 0.7354 | 0.7366 | 0.7324 | 0.7418 |

## B.3 SIMILARITY SEARCH

Results for GRACE, GCA, BGRL, and AFGRL are from Lee et al. (2022).

**Table 6:** Performance on similarity search measured by Sim@5 and Sim@10.

| | | GRACE | GCA | BGRL | AFGRL | SGRL-200 | Ours-SGRL |
|---|---|---|---|---|---|---|---|
| WikiCS | Sim@5 | 0.7754 | 0.7786 | 0.7739 | 0.7811 | 0.7968 | 0.7967 |
| | Sim@10 | 0.7645 | 0.7673 | 0.7617 | 0.7660 | 0.7825 | 0.7825 |
| Computers | Sim@5 | 0.8738 | 0.8826 | 0.8947 | 0.8966 | 0.8883 | 0.8900 |
| | Sim@10 | 0.8643 | 0.8742 | 0.8855 | 0.8890 | 0.8785 | 0.8809 |
| Photo | Sim@5 | 0.9155 | 0.9112 | 0.9245 | 0.9236 | 0.9186 | 0.9200 |
| | Sim@10 | 0.9106 | 0.9052 | 0.9195 | 0.9173 | 0.9115 | 0.9136 |
| Co.CS | Sim@5 | 0.9104 | 0.9126 | 0.9112 | 0.9180 | 0.9036 | 0.9071 |
| | Sim@10 | 0.9059 | 0.9100 | 0.9086 | 0.9142 | 0.8971 | 0.9020 |
| Co.Physics | Sim@5 | - | - | 0.9504 | 0.9525 | 0.9523 | 0.9518 |
| | Sim@10 | - | - | 0.9464 | 0.9486 | 0.9480 | 0.9474 |

## C RELATED WORK

This work relates closely to several areas of machine learning, including proximity-preserving graph embedding, measures of distortion, and general minimum distortion embedding (not limited to graphs).

Some well-known examples of proximity-preserving graph embedding methods include DeepWalk (Perozzi et al., 2014), Node2Vec (Grover & Leskovec, 2016), and LINE (Tang et al., 2015). DeepWalk is a scalable method for learning node embeddings by combining random walks with the Word2Vec skip-gram model. It treats random walks as sequences analogous to sentences in natural language processing, capturing local graph structure through node co-occurrences. The method produces continuous vector representations that preserve social and structural relationships and is a pioneer for proximity-preserving embedding. Node2Vec extends DeepWalk by introducing a flexible, biased random walk strategy that balances local and global exploration of graph neighborhoods, controlled by parameters $p$ and $q$. This allows Node2Vec to capture diverse connectivity patterns. LINE is a pioneering approach for embedding large-scale information networks into low-dimensional vector spaces while preserving both first-order and second-order proximity, which has influenced later proximity-preserving embedding methods, such as HOPE (Ou et al., 2016) and APP (Zhou et al.,

2017). While DeepWalk, Node2Vec, and LINE only preserve symmetric proximities, APP preserves asymmetric proximity. HOPE, on the other hand, preserves asymmetric transitivity.

Unlike all previous work, we regulate distortion directly through the loss function. Since $\mathcal{D}_\rho$ measures pairwise distance deviations for any pair, with matching distance measures, it can preserve the $k$th-order proximity for any $k \geq 1$ and covers both symmetric and asymmetric proximity.

Literature presents a wide range of distortion measures, some based on differences between two distances (Nowak et al., 2024), and some based on ratios of two distances (Abraham et al., 2005; 2007; 2011; Vankadara & von Luxburg, 2018). Similar to the $\sigma$-distortion in Vankadara & von Luxburg (2018), our distortion measure $\mathcal{D}_\rho$ is also based on the ratio of two distances, but it uses normalized distances, so only the relative change matters. In contrast, $\sigma$-distortion directly uses the unnormalized distances; therefore, large ratios dominate the small ratios. Other notable work includes WEmbed (Ma et al., 2024), which leverages a weighted distortion measure inspired by hyperbolic geometry to preserve complex signed graph structures.

In contrast, some methods employ embedding algorithms that indirectly minimize distortion. For example, curvature regularization is introduced in Pei et al. (2020) to reduce distortion in proximity-preserving node embedding. In Pei et al. (2020), distortion is defined as the distance divergence (ratio) between an embedding manifold and its ambient Euclidean space. Isomap (Tenenbaum et al., 2000) is a novel algorithm for nonlinear dimensionality reduction that preserves the intrinsic geometry of data on a low-dimensional manifold by using geodesic distances estimated through a nearest-neighbor graph, extending classical multidimensional scaling (MDS) to capture nonlinear structures effectively. Poincaré embedding was introduced in Nickel & Kiela (2017) for embedding hierarchical graph structures within the Poincaré ball model of hyperbolic space, leveraging its geometry to preserve tree-like and complex network properties with low distortion, directly addressing distortion in non-Euclidean spaces.

Directly finding embeddings with minimum distortion is also pursued in mathematical programming. The term "minimum-distortion embedding" (MDE) was first formalized in Agrawal et al. (2021) as a constrained optimization problem, which can incorporate various distortion functions. Distortion functions are all functions of Euclidean distances. It was shown that only in a few special cases can MDE problems be solved exactly; for other cases, they can only be approximated by using a projected quasi-Newton method. In this paper, we directly minimize distortion in the learning objective for various graph learning tasks, and use distortion as a regularization term in the loss function, whereas in Agrawal et al. (2021), selecting embeddings involves finding an exact or approximate solution to the MDE problem, implemented within a mathematical programming framework.

## D  THE USE OF LARGE LANGUAGE MODELS (LLMS)

The paper is written with assistance from LLMs. We use LLMs to find information such as LaTeX commands and symbols, Python packages and library functions.

