# OpenReview forum: "Distortion Regularization for Distance-Preserving Graph Embedding"
_ICLR.cc/2026/Conference — ICLR 2026 Conference Withdrawn Submission_

### Official Review · Reviewer_kf7v · 2025-10-31

**Soundness:** 2
**Presentation:** 3
**Contribution:** 2
**Rating:** 4
**Confidence:** 4

**Summary:**

This paper introduces a general, scale-invariant distortion measure that can be added as a regularization term to existing graph embedding loss functions to better preserve pairwise distances. The authors theoretically demonstrate that Gaussian kernel embedding is a specific type of minimum-distortion embedding and compare the relative strictness of their proposed measure against other metrics like Spearman's footrule and MAP. Experiments show that adding this distortion regularization to models like GCN and SGRL marginally improves performance on downstream node classification, clustering, similarity search, and link prediction tasks across benchmark datasets.

**Strengths:**

1. The paper addresses a fundamental problem in machine learning: distance preserving representation learning. The formal analysis presented in the paper (Sections 3, 4 and 5) is generally easy to follow and rigorous, with a couple of exceptions as mentioned later.

2. The authors have applied the proposed distortion regularization in several graph related downstream tasks. This aspect of research in the graph domain is highly under explored.

3. Concepts like optimality of distortion measures (embedding f with zero distortion under a specific distortion measure ensures
strict distance preserving) and strictness of distortion measures are very important for learning good self-supervised (or unsupervised) representations. The paper does a decent job in presenting them.

**Weaknesses:**

1. The originality of the theoretical analysis is not clear in the paper. How many of the proposed concepts are novel in the paper and how many of them are adapted from the related literature? The paper hardly cites any related works in the core technical sections of it (i.e., Sections 3, 4 and 5). However, some of these concepts (for e.g., scale invariance) are taken from "Measures of distortion for machine learning" (NeurIPS 2018). The authors should clarify that and cite related works wherever applicable. There is another paper which addresses the preservation of distance in the graph embedding - "A framework to preserve distance-based graph properties in network embedding" (Soc. Netw. Anal. Min. 2022). The authors should look into them carefully.

2. The core technical sections of the paper does not involve graph at all. All the metrics used in the analysis of distortion are general metrics. However, the paper uses graph in its title and in the experiments. If the analysis is quite generic in nature, what was the need to do the experiments only for the graph downstream tasks?

3. The improvements in the experimental results are too marginal in nature for all the downstream applications selected. Does preserving distance in the embedding space not really helpful in solving the downstream applications? What was the additional computational burden to add the distortion regularizer in the loss?

**Questions:**

Several clarifications needed in the paper.

1. Apart from scale invariance, what other axioms a good distortion function should satisfy?

2. What is the purpose of defining distortion function and distortion loss separately in Equations 2 and 3? Can't you just minimize the distortion function directly? Similarly, what is the difference between Equations 14 and 15?

3. In Section 4, can you first formally define what a "Minimum Distortion Embedding" is? It is informally explained later, but not formally defined.

4. Equation 15 is not clear to me.

5. In Definition 2 in Section 5.2, please formally define "Optimality in A".

6. In Section 6.1.1, what are the first and second stages for node classification and which model parameters are you updating with the cross entropy loss?

7. SGLR is introduced in L364 without the full form and without any citation. If it is Scattering Graph Representation Learning, it should be clarified and cited appropriately.

---

### Official Review · Reviewer_LVP5 · 2025-11-01

**Soundness:** 3
**Presentation:** 3
**Contribution:** 2
**Rating:** 4
**Confidence:** 5

**Summary:**

This paper proposes distortion regularization for graph embedding. The authors define a differentiable, scale-invariant distortion measure​ that penalizes deviations between pairwise distances in the original graph and the embedding space. The distortion term can be added to any graph-learning objective (e.g., GCN, SGRL, GAE) as a regularizer. The paper also shows that Gaussian kernel embedding is a form of minimum-distortion embedding. Empirically, adding the distortion term improves performance on node classification, clustering, similarity search, and link prediction benchmarks.

**Strengths:**

* Clear formulation: The paper provides a mathematically sound definition of distortion, proves scale-invariance, and connects the proposed metric to Gaussian kernel embeddings. The idea of directly regularizing pairwise distance preservation is simple, broadly applicable, and relevant to geometric representation learning.
* Plug-and-play regularizer: The distortion loss is generic and can be applied to many graph encoders without architectural changes.
* Empirical coverage: Experiments span both supervised and unsupervised settings, and consistent (if small) performance gains are reported across diverse tasks.

**Weaknesses:**

*  Scalability: The pairwise loss is $O(n^2)$ in the number of nodes $n$; the paper does not discuss computational implications or sampling strategies for large graphs.
* Limited intuition: The geometric intuition (why distortion correlates with better performance) could be elaborated.
* Missing ablations and baselines:
   - Regularization applied only on the final layer: The paper applies the loss once on the final embeddings, not across intermediate layers, so it is uncertain how well distance preservation holds throughout the network or whether earlier representations are still distorted, or if that has an impact.
   - No comparison to classical distance-preserving embeddings: Baselines include GCN/SGRL/GAE variants, but not classical geometric methods (e.g., Laplacian Eigenmaps, Isomap, MDS, or direct Minimum-Distortion Embedding solvers). Without these, it is difficult to compare the learned embeddings to purely distance-preserving solutions.
  - No distortion-only embeddings: The authors argue that “minimum distortion is a fundamental principle for graph embedding,” but they never train or evaluate embeddings using only the regularization term (without task losses). It is thus unclear whether the distortion objective itself can yield meaningful structure-preserving embeddings that are useful for downstream tasks, or if it merely serves as a mild de-collapsing term or serves another purpose that aids existing models.
* Limited analysis of the embeddings: The core claim is that distortion regularization leads to “minimum-distortion” embeddings, yet the paper never reports actual distortion values of learned embeddings, nor any correlation between distortion and downstream accuracy. Improvements are shown only through task metrics.

**Questions:**

1. Can you report average distortion values (before and after regularization) or show that lower correlates with higher accuracy?
2. What happens if you train using the regularization term alone. Do the resulting embeddings remain useful for downstream tasks?
3. How do you handle computational cost for large number of pairwise terms?

---

### Official Review · Reviewer_WWrE · 2025-11-03

**Soundness:** 3
**Presentation:** 3
**Contribution:** 2
**Rating:** 4
**Confidence:** 4

**Summary:**

This paper proposes a distortion regularization term for linearly distance-preserving graph embedding. The key idea is to introduce a general distortion measure that can be incorporated into loss functions to maintain linear pairwise distance relationships during graph embedding.  The authors provide theoretical analysis on scale invariance and strictness, and evaluate the approach across multiple downstream tasks, including node classification, clustering, similarity search, and link prediction. The paper is mathematically well-structured and addresses an important aspect of graph representation learning, namely distance preservation. However, while the proposed framework is conceptually solid, the novelty and empirical depth remain limited

**Strengths:**

S1. The paper provides a clear formalization of “distortion” in the context of graph embedding, and unifies multiple prior intuitions under a single regularization view.

S2. In principle, the proposed distortion regularization can be applied to different graph learning architectures and objectives without major modification.

S3. The formal definitions, such as scale-invariance and strictness of distortion measures, are mathematically sound and well-structured.

S4. Across multiple benchmark tasks, the regularization consistently yields slight improvements over baseline models, suggesting some robustness.

**Weaknesses:**

W1. It is not clear what the difference is between the proposed method and existing regularizations, such as curvature regularization or contrastive embedding constraints.

W2. The paper shows improved metrics but provides little insight or analysis on why the proposed distortion regularization is effective and how it specifically prevents distortion.

W3. The claim that distortion regularization is a “fundamental principle” seems overstated, as the evidence mainly shows incremental improvement on known benchmarks.

**Questions:**

1. Could the authors clearly explain how the proposed distortion measure differs from existing curvature- or distance-preserving regularizations?

2. What is the computational overhead of adding distortion regularization, and how does it scale with graph size?

3. Can the authors provide visualizations of embeddings before and after distortion regularization to illustrate its actual impact?

---

### Official Review · Reviewer_evik · 2025-11-09

**Soundness:** 2
**Presentation:** 2
**Contribution:** 2
**Rating:** 4
**Confidence:** 5

**Summary:**

This paper studies distance preservation in graph embedding and introduces a generic differentiable distortion measure that can be directly integrated into loss functions as a regularization term. The distortion formulation is scale invariant and discourages embedding collapse by pushing node representations apart while maintaining relative pairwise distances. The authors theoretically analyze the strictness of different distortion metrics and prove that Gaussian kernel embedding corresponds to a form of minimum distortion embedding. Experiments across multiple node and link level tasks show that adding distortion regularization consistently improves performance over strong graph representation baselines on standard datasets, particularly for classification and link prediction.

**Strengths:**

The following are 3 strong points of this paper:
1. The proposed distortion loss is independent of model architecture and can be incorporated into many existing graph learning pipelines, making the contribution broadly useful.

2. The paper formally proves scale invariance (Theorem 1) and provides analysis of the strictness hierarchy among distortion measures (Theorems 3 and 4), which strengthens rigor.

3. The proposed distortion regularization improves performance across different tasks such as node classification,clustering, similarity search, etc.

**Weaknesses:**

The following are 3 weak points of this paper:

1. My first major concern is on the computational scalability aspects. In particular, the proposed distortion loss uses all pairwise node distances, yielding quadratic complexity.. while it is feasible for benchmark datasets, this is not addressed for large-scale graphs.

2. As distortion implicitly spreads embeddings, tasks that depend on local structural compactness or community detectability might degrade, but no negative results or caveats are explored.

3. All datasets considered in this paper are classical citation/product/coauthor benchmarks.

**Questions:**

(a) Please provide your responses for the above 3 weak points...

(b) Can you please provide failure case analysis by showing situations where distortion regularization may hurt (e.g., hierarchical graphs, scale-free graphs).

(c) Author stated that dS and dH are application-dependent. Accordingly, can you do some experiments by varying cosine vs. Euclidean vs. shortest path.. this would clarify robustness as well.

---

### Note · Authors · 2025-11-23

I have read and agree with the venue's withdrawal policy on behalf of myself and my co-authors.